# Experimental Analysis and Evaluation of Automatic Control System for Evenly Scattering Crushed Straw

Bokai Wang [1], Feng Wu [1], Fengwei Gu [1,*], Hongchen Yang [1], Huichang Wu [1] and Zhichao Hu [2,*]

1 Nanjing Institute of Agricultural Mechanization, Ministry of Agriculture and Rural Affairs, Nanjing 210014, China
2 Graduate School of Chinese Academy of Agricultural Sciences, Beijing 100083, China
* Correspondence: gufengwei@caas.cn (F.G.); huzhichao@caas.cn (Z.H.)

**Abstract:** In order to improve the solution to the unachieved uniformity of straw throwing, the unachieved qualified rate of coverage and the uneven straw throwing in sowing wheat without a tillage process after the rice harvest, and to change this unsatisfied quality of the straw mulch, a set of automatic control systems for straw throwing and covering was designed innovatively. An STM32 microcontroller was used as the main control unit, and the torque-acquisition system was used to collect the torque of the cutter roller shaft in real time and convert it into the conveying signal of the crushed straw. The control system changes the conveying quantity of broken straw in real time, through the dynamic response. This process realizes the optimal dynamic matching between the conveying amount of crushed straw and the impeller speed. We set up two kinds of tests: a straw-crushing-and-throwing system test bench (T6)6 with an automatic control system and a control test bench (C) without an automatic control system. T1 to T5 are, in turn, 0.85 m/s, 1.0 m/s, 1.15 m/s, 1.30 m/s and 1.45 m/s. For the C test, six test levels of 0.85 m/s (C1), 1.0 m/s (C2), 1.15 m/s (C3), 1.30 m/s (C4), 1.45 m/s (C5) and variable speed test (C6) were also set as control tests. The running time of the test-bed at each test level was 10 s; taking the throwing uniformity of the crushed straw and the rate of coverage as indexes, the rapid effect of the throwing-impeller speed on the test indexes at six levels was studied, and compared with the control test. Based on the great practical needs of this problem, this experiment innovatively realized the automatic regulation of the rotating speed of the scattering impeller at different forward speeds. Although some experimental innovations have been made in this study, the smashing knife (group) of the knife roller shaft will hit the ground during the rotation, which brings uncertainty and certain experimental errors to the real-time monitoring of the torque signals. In the next step, more sensors and intelligent algorithms will be added to the system, to reduce the knife throwing.

**Keywords:** crushed straw; no-tillage sowing; coverage uniformity; coverage qualification rate; throwing-impeller speed; automatic regulation





## 1. Introduction

The no-tillage sowing mode in straw-mulched land is developing rapidly in central China [1,2]. It has great advantages compared with traditional tillage methods [3,4].

The technical core of no-tillage sowing in straw mulch is the mechanized-straw-crushing and throwing system, which mainly depends on the comprehensive action of cutting force, picking thrust and high-speed air flow generated by the high-speed rotating knife, set to crush and throw materials. It has the characteristics of a good crushing effect and strong conveying capacity, and is widely used in various forage harvesters and straw-crushing and returning machines in China at present [5]. However, we found many problems in experiments from 2016 to 2021, such as an unsatisfied uniformity of straw scattering and low-unsatisfied rate of coverage, due to the fluctuation of the feed rate. The fundamental reason was that the speed of the scattering component at the tail of the no-till

sowing equipment was fixed, while the walking speed of the no-till sowing equipment in the field was variable. Moreover, the quantity and weight of the rice-straw-per-unit area left in the same plot or different plots were also different. Even when the no-till sowing machine operated at the same speed, the amount of straw it treated would change in real time, eventually resulting in the density and quantity of crushed straw ejected from the channel changing all the time. Therefore, it is easy to cause the problem of unsatisfied uniformity of straw throwing and low-quality rate of coverage. This problem has aroused widespread concern among Chinese farmers and the news media, because the qualified rate of coverage and uniformity of straw throwing directly affect the operation of water and fertilizer and the utilization of light and heat [6,7].

Wang et al., set up a no-tillage sowing test bed to evaluate the coverage uniformity of rice straw after throwing; however, the impeller speed of the test bed was fixed under different working conditions, and its adaptability to the fluctuation of the straw-feeding amount was unsatisfied [8]. Zhai improved the throwing effect to some extent, but all kinds of structure and motion parameters were still fixed [9]. Qin et al., adopted a theoretical analysis and the ADAMS simulation method to design the structure of the impeller, and determined the optimal structural parameters of key parts such as the stubble cutter, which was helpful in improving the sowing uniformity [10]. Yuan et al., carried out mechanical and modal analyses on the impeller device for crushing and throwing corn stalks, using a discrete element to study the throwing of the straws, and achieved some result [11]; Zhang et al., designed a machine which realized the adjustment of straw-covering uniformity and distance by manually adjusting the spacing of the guide-vane groups on the guide plate, but it still did not realize automatic control [12]. Wang. et al., designed a device to improve the quality of crushing and returning to the field, by adjusting the parameters of the fixed knife in real time [13].

The current progress can be divided into two directions. One is to simulate the movement law of crushed straw in the mechanical system by numerical calculation (analysis), obtaining the best parameters by optimizing the structure and movement parameters, so as to improve the throwing quality and the effect of the straw. The second is simple structural parameters research. The researchers have made important achievements, but the results of these two kinds of study still fix the structural parameters but cannot adjust the working parameters in real time, according to the change in flow and quantity of crushed straw. Moreover, these achievements focus on the "intermediate links", such as straw crushing and straw conveying, while the control system of the straw-throwing impeller is the "last key link".

Focusing on the above problems, this paper designed a set of automatic control systems for straw throwing, based on the test bed. The closed-loop control of the DC brushless motor is carried out by comparing the real-time rotation speed with the theoretical rotation speed, so as to realize the adaptive regulation of the rotation speed of the straw-throwing impeller at different straw-ejection amounts, and to ensure the quality of broken-straw coverage.

## 2. Materials and Methods

### 2.1. Straw-Crushing-and-Throwing System Test Platform

#### 2.1.1. Working Principle

The crushing-and-throwing test bed independently developed by the research group was selected for the experiment. The test bed is powered by the tractor PTO, and the throwing-impeller and the throwing device (tail part) are made to work by the deceleration mechanism and the transmission mechanism, respectively. The equipment is shown in Figure 1, the throwing principle of the crushed straw is shown in Figure 2, the main performance parameters is shown in Table 1.

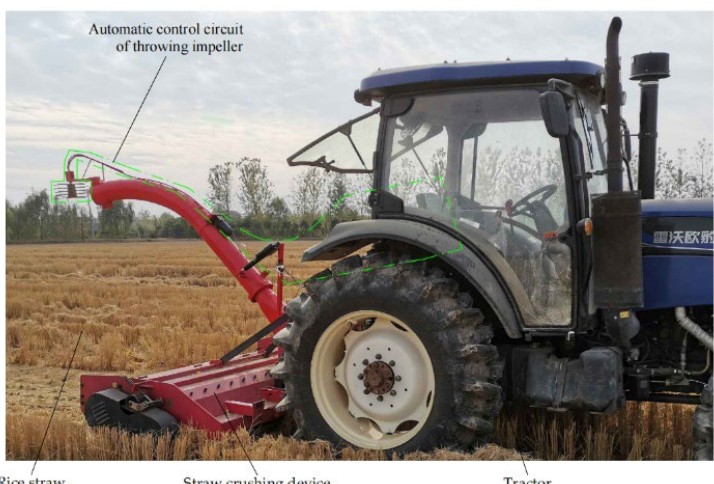

**Figure 1.** Overall structure of test bed.

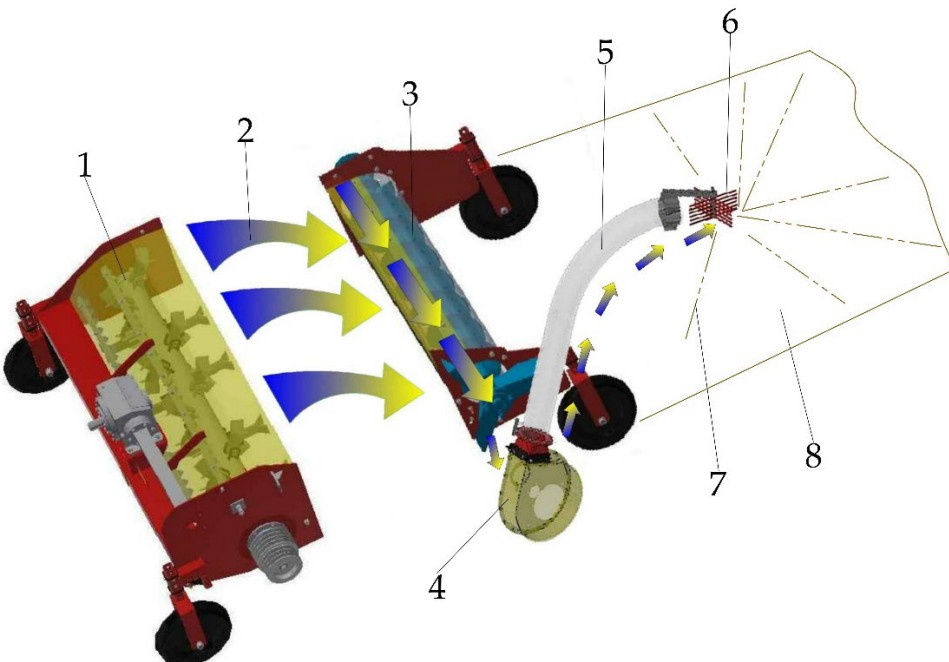

**Figure 2.** Schematic diagram of crushed-straw throwing. 1. Straw-crushing device. 2. Movement direction of crushed straw. 3. Spiral-conveying device. 4. Fan. 5. Straw-crushing channel. 6. Throwing impeller. 7. Straw-crushing trajectory. 8. Straw-crushing area.

**Table 1.** Main performance parameters.

| Parameter | Numerical Value |
|---|---|
| Test bed mass/kg | 1480 |
| Matching power/Kw | 70~90 |
| Roller speed/(rmin$^{-1}$) | 2200 |
| Working width/m | 2.2 |
| Throwing-impeller speed/(rmin$^{-1}$) | 500~1200 |

2.1.2. Automatic Control System for Straw Crushing and Throwing

Figure 3 shows the basic principles of the automatic control system for the uniform scattering of crushed straw, which include: ① a 32-bit microprocessor STM32F103RCT6, which is used as the data-processing hardware platform, with a CPU processing speed of

72 MHz; ② 57BL95S15-230 DC brushless motor with rated speed of 3000 r/min; torque is 0.5 N·m; ③ ZM-6615 low-voltage DC brushless motor driver (18 to 60 V) ④12 V batteries (2); ⑤ TB1808-N2 inductive-proximity sensors (2 sensors, with a detection distance of 0~8 mm and a detection frequency of 0~1000 HZ); ⑥ E50S8-2400-3-T-24 encoder (maximum allowable speed: 5000 rpm); ⑦ Laptop, equipped with driver and data-acquisition program.

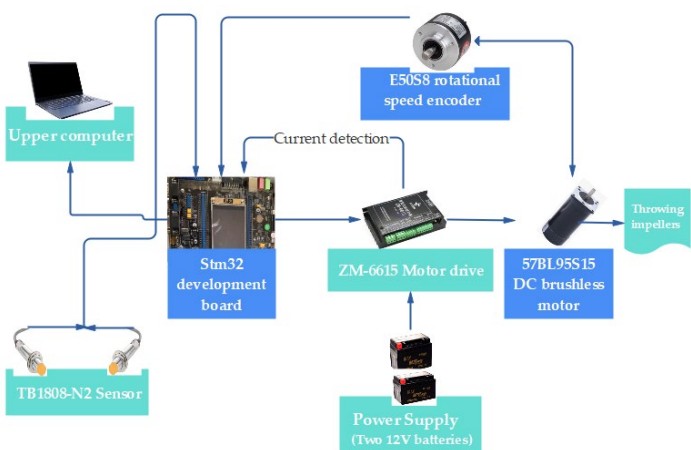

**Figure 3.** Automatic control system for uniform scattering of crushed straw.

When working, two TB1808-N2 inductive-proximity sensors installed at both ends of the cutter roller shaft and the matching detection disk are selected as detection devices to collect the torque information from the cutter roller shaft in real time, and then the information is transmitted to the microprocessor through the serial port. After the microprocessor has interpreted and processed the information, it outputs PWM signals with the corresponding duty ratios. The PWM signal regulates the motor speed, and the brushless motor drives the flexible shaft to change the rotation speed of the throwing impeller. Meanwhile, the encoder collects the rotation-impeller speed, and closed-loop control of the DC brushless motor is carried out by comparing the real-time rotation speed with the theoretical rotation speed, so as to realize the self-adaptive control of the rotation speed of the straw-throwing impeller in different straw-ejection amounts and to improve the uniformity of the straw throwing and the qualified rate of coverage, thus ensuring the coverage quality of the crushed straw.

The real-time monitoring of the torque signal determines the impeller speed, and the torque acquisition is very important. Figure 4 shows the installation position of the proximity sensor. When the system works, the phase difference between the two TB1808-N2 inductive-proximity sensors installed at both ends of the cutter roller shaft and the matching detection disk reflects the twist angle of the cutter roller shaft, and it can be seen from the material mechanics that there is a relationship [14]:

$$T_1 = \frac{\pi d^4 G \alpha}{32L} \tag{1}$$

here, $T_1$ is the torque (N·m) between two sections of cutter roller shaft, (N·m); $d$ is the roller shaft diameter, m; $G$ is the shear modulus, pa; $\alpha$ is the twist angle, rad; and $l$ is the distance between the two sections, m.

The phase difference is converted into the torque signal in real time through different torsion angles, and, according to a large number of experimental data in the early stage, the optimal mathematical model between the torque signal and delivery quantity is established in the control system; the optimal mathematical model between different delivery quantities and the throwing-impeller speed is also established. Through these two mathematical models, the real-time torque signal is converted into the optimal throwing-impeller speed, and the optimal dynamic matching of the throwing-impeller speed is completed.

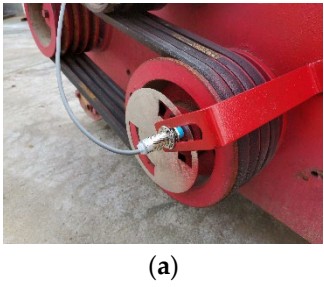
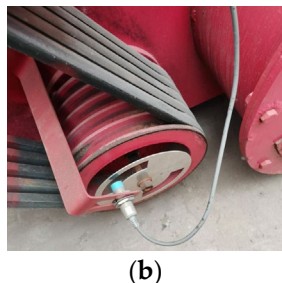

(**a**)        (**b**)

**Figure 4.** Installation diagram of proximity sensor. (**a**) left side; (**b**) right side.

### 2.1.3. Automatic Control System Software

Figure 5 shows the flow chart of the automatic control system for the uniform scattering of crushed straw. An automatic system is designed according to the principle of straw scattering under different working conditions. Firstly, the torque of the cutter roller shaft is detected, and then the corresponding control method is adopted for the throwing impeller. Different rotational speed states of the throwing impeller correspond to different action states of the inductive-proximity sensor, and different switching signals are input to the I/O pin of the controller. The depth-PID-control algorithm is used to adjust the impeller speed. Figure 6 shows a block diagram of the automatic control mechanism of straw throwing based on depth PID control. The controller calculates the theoretical rotation speed and real-time roller speed shaft according to the pulse numbers recorded and stored by TIM2, TIM3 and TIM4, and uses them as the input for the incremental depth-PID-control algorithm to calculate and output PWM signals to control the speed regulation of the brushless DC motor [15,16].

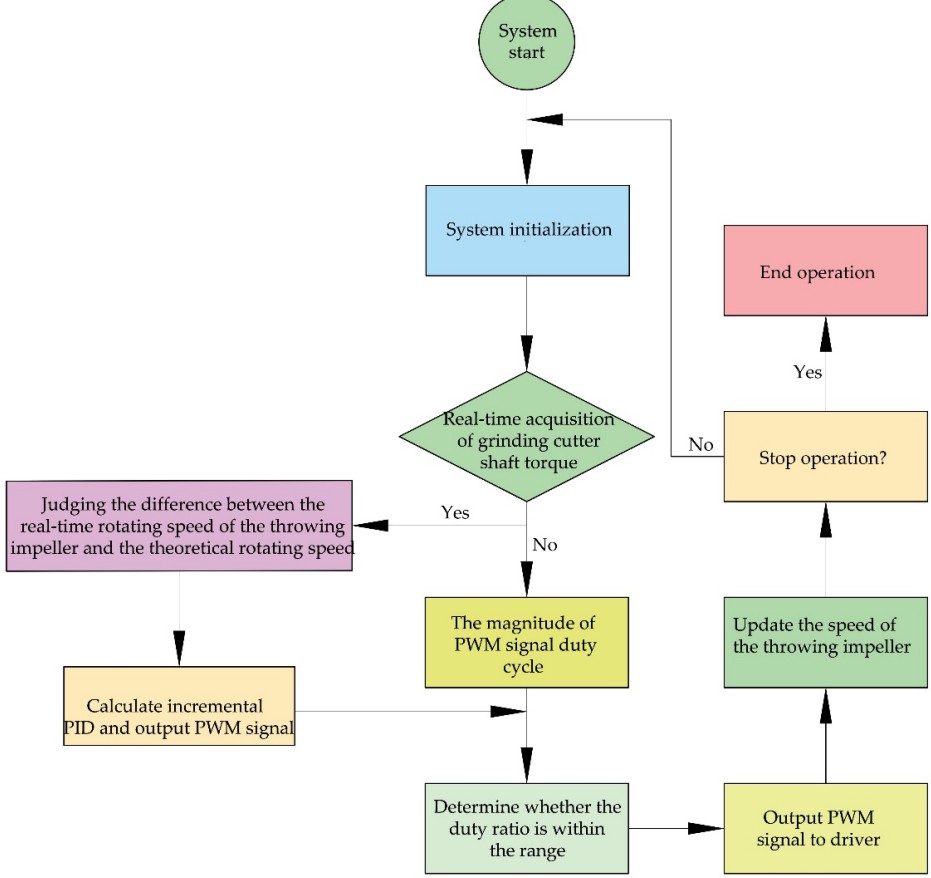

**Figure 5.** Program flow chart of automatic control system for straw covered after throwing.

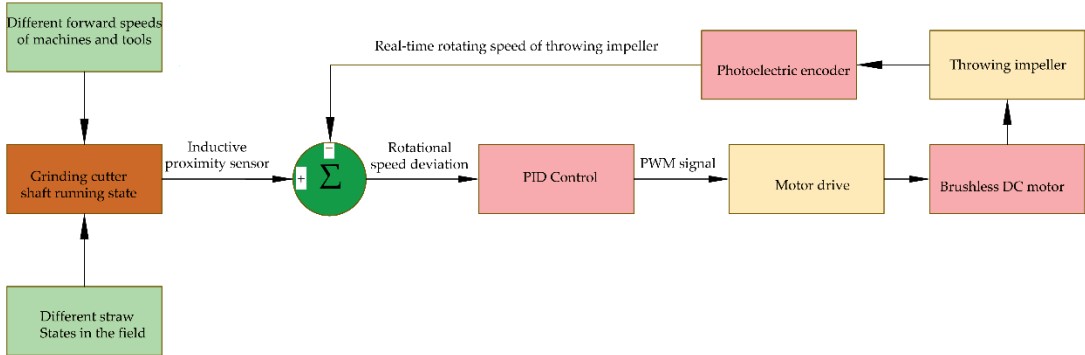

**Figure 6.** Automatic control mechanism of straw throwing based on depth PID control.

Through data analysis and the processing of rectangular-wave-signal input by the PA0, the control system obtains the high-level duration, T, of the input signal, which reflects the torque of the cutter roller shaft; the control system then substitutes the average T value of the three rotation periods to calculate the corresponding value, and compares this value with the value obtained in the initialization or the previous period, to determine whether it is necessary to adjust it. When the result is affirmative, the timer value is configured to change the PWM duty ratio, the real-time adjustment of the impeller speed is realized, and the conveying amount and throwing of the impeller are completed.

*2.2. Depth-PID-Parameter Verification Based on Simulink*

In Simulink, the proportional, integral and differential coefficients are changed in turn for simulation (Figure 7), so as to obtain a better combination of the PID control parameters and provide a reference for the actual tuning of the PID parameters [17]. First, set different proportional coefficients, $K_p$, keep the integral coefficient $K_i$ and differential coefficient $K_d$ at 0, and input a step-response signal with an amplitude of 1 to obtain a step-response curve, as shown in Figure 7a. $K_p$ increases, the response speed changes a little, the response steady-state value increases, the steady-state error decreases, the rise time decreases, and the overshoot increases. When $K_p > 0.35$, the overshoot increases sharply. Compared with $K_p = 0.35$, $K_p = 0.25$ has a similar steady-state error and relatively smaller overshoot, so $K_p = 0.35$ is selected after comprehensive consideration. Secondly, set different integral coefficients for Ki, keep the proportional coefficient $K_p$ and differential coefficient $K_d$ at 0.35 and 0, respectively, and input a step-response signal with an amplitude of 1, to obtain a step-response curve (Figure 7b). With $K_i$ added, the response speed increases, and with the increase in $K_i$, the response steady-state value increases, the steady-state error is basically eliminated, the rising time is basically unchanged, and the overshoot slightly increases, compared with that before $K_i$ is added. When $K_i < 0.25$, the oscillation frequency of the system response is high, near the steady-state amplitude; when $K_i = 0.25$, the system response is relatively stable, but the oscillation around the amplitude is small; when $K_i > 0.25$, the overshoot increases sharply, taking into consideration the comprehensive selection of $K_i = 0.25$. Finally, set differential gain coefficients, $K_d$, respectively, and input a step-response signal with an amplitude of 1 to obtain a step-response curve, as shown in Figure 7c. When $K_p$ is added, the response speed increases, and with a decrease in $K_d$, the overshoot decreases. The change in $K_d$ has little influence on steady-state error and rise time; when $K_d > 0.009$, the oscillation frequency of the system response is high, near the steady-state amplitude, while when $K_d = 0.009$, the system response is relatively stable and the oscillation is small, near the steady-state amplitude. Comprehensive consideration is given to $K_d = 0.009$. Change the input signal of the control model to a sine wave signal, as shown in Formula (11), and obtain the incremental-depth-PID sine-response curve, as shown in Figure 7. The control system can respond quickly, according to the change of signal. At the beginning, the output signal fluctuates greatly, and the error between the output signal and the input signal is large. Then, after gradually decreasing for 0.2 s, the

output signal can change quickly with the input signal, and the error is basically kept within 0.02, with high control accuracy and good stability. Finally, the PID parameters are $K_p = 0.35$, $K_i = 0.25$ and $K_d = 0.009$.

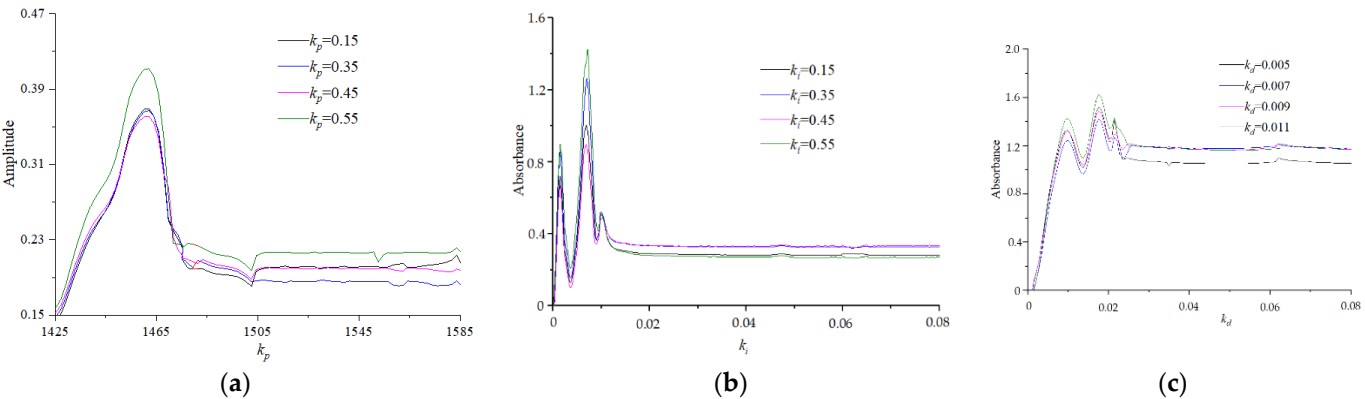

**Figure 7.** Depth-PID-parameter verification at low speed: (**a**) Step response curves under different values of $K_p$; (**b**) Step response curves under different values of $K_i$; (**c**) Step response curves under different values of $K_d$.

### 2.3. Experimental Design

The test refers to "Sower in clean area with whole straw stubble", NY/T1768-2009 "Technical specification for quality evaluation of no-tillage sower", and "Straw crushing and returning machine" [18]. The total amount of straw stubble after the rice harvest is selected as the field, and the average moisture of the straw is 44%.

There were 40 rectangular test areas of 2.2 m × 90 m in the test area, and the starting line was set at the 10 m of the test area. The automatic control system was 0.85 m/s (T1), 1.0 m/s (T2), 1.15 m/s (T3), 1.30 m/s (T4) and 1.45 m/s (T2). The control test bed (CK) without the automatic control system was set, and six test levels of 0.85 m/s (C1), 1.0 m/s (C2), 1.15 m/s (C3), 1.30 m/s (C4), 1.45 m/s (C5) and a variable speed test (C6) were correspondingly set, with two changed.

During the test, the test site of each test level was randomly selected from 40 test areas; in the straw-crushing-and-throwing test with an automatic control system, the test bed advance for 10 s at each level. Every 0.25 s was taken as a time step, and the average speed signal of each time step was taken as a speed point. There were 40 data points in each test level, and the throwing uniformity of crushed straw and rate of coverage were taken as indexes. The rapid effect of the throwing-impeller speed on test indexes in each test level of the two test treatments was studied, and the results were compared.

During the test, the cutter roller shaft always operated at the maximum rotation speed of 2200 r/min; when there was no control system, the impeller speed was kept at the set value of 800 r/min; when there was a control system, the impeller speed could be automatically adjusted, and the speed-adjustment range was 500~1200 r/min. After the test was completed, the index of the scattered area of the crushed straw was obtained.

### 2.3.1. Detection Method of Scattering Unevenness

After each level of the two test beds was finished, the test beds stopped moving forward. In the corresponding test areas, the areas were divided according to the initial line of the machine test and the time step (0.25 s), in turn. The test areas after each test level were divided into 40 areas on average, and each area was divided into 10 districts on average (Figure 5). The scattering unevenness of each district was obtained. Each district

was measured using a 5-point sampling method (Figure 8) and the straw was picked up manually [19,20].

$$M_{zj} = \frac{\sum\limits_{y=1}^{5} m_{jy}}{5} \tag{2}$$

$$M_a = \frac{\sum\limits_{j=1}^{10} M_{zj}}{10} \tag{3}$$

$$F_b = \frac{1}{M_a} \sqrt{\frac{\sum\limits_{j=1}^{10} \left(M_{zj} - M_a\right)^2}{9}} \times 100 \tag{4}$$

here, $M_{zj}$ is the average mass of straw at the jth measuring point, $g^2$; $M_{jy}$ is the quality of straw in the $j$ sampling plot, g; $M_a$ is the average mass of straw, g; $F_b$ is the uneven distribution of straw in the area, by %.

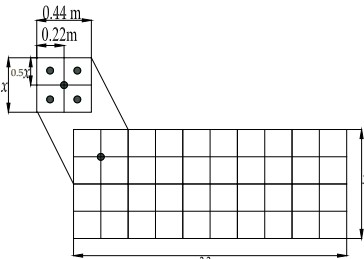

**Figure 8.** Scattering-unevenness test.

Within the working width, the uniformity of the transverse distribution of crushed straw along the width. The more evenly the crushed straw is scattered in the whole width after the operation, the better the operational effect [21,22].

### 2.3.2. Detection Method of Qualified Rate of Coverage

The weight of the straw in the unqualified throwing area at each time step of each test level is measured. The unqualified throwing area refers to the extension of 0.8 m to the left, and 0.8 m to the right, of each test area. Figure 9 shows the range of the measured coverage; the qualified coverage rate is [23,24].

$$P_j = \frac{m_{jl} + m_{jr}}{m_{jl} + m_{jr} + m_j} \times 100\% \tag{5}$$

here, $P_j$ is the qualified rate of the coverage of the jth survey area, by %; $m_{jl}$ is the mass of straw in the left unqualified area, g; $M_{jr}$ is the mass of straw in the right unqualified area, g; and $M_j$ is the mass of straw in the middle unqualified area (qualified area).

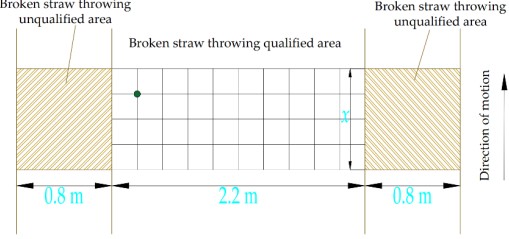

**Figure 9.** Detection method of qualified rate of coverage.

## 3. Results

### 3.1. Speed Response of Different Forward Speeds

3.1.1. Detection Method of Qualified Rate of Coverage

Figure 10a–e show the changes in straw-scattering uniformity in six levels under the T and C treatments. The uniformity of straw scattering corresponding to T1, T2, T3, T4 and T5 was 86.97%, 89.98%, 90.75%, 91.49% and 89.06% respectively. The average evenness of straw scattering under the five treatments of C was 75.23%, 82.12%, 82.42%, 76.27% and 71.41%, respectively. The average evenness and fluctuation of straw scattering at five levels under the two treatments of T and C were random; they increased and then decreased. The difference was that, in the T5 treatments, the corresponding average throwing uniformity of the straw randomly increased, within a high range of 86.97~91.49%, and the variation range was small (less than 5%), which indicated that the automatic control system could monitor the torque of the cutter roller in real time, and could adjust the rotating speed of the throwing impeller in the best turntable, according to the torque change of the cutter roller in the case of increasing feed rate, thus keeping the throwing uniformity of the straw within a high range. However, under the five treatments of C, the average throwing uniformity of the straw was random, and in a low range, with the increase in advancing speed of 71.41~82.42%, and the range of change as high as 11%. The throwing uniformity moved upward slightly and then dropped sharply with a movement of advancing speed, which showed that the throwing impeller without the automatic control system was not suitable for different advancing-speed conditions; the impeller speed was less than 800 r/min, with the amount of crushed straw being thrown.

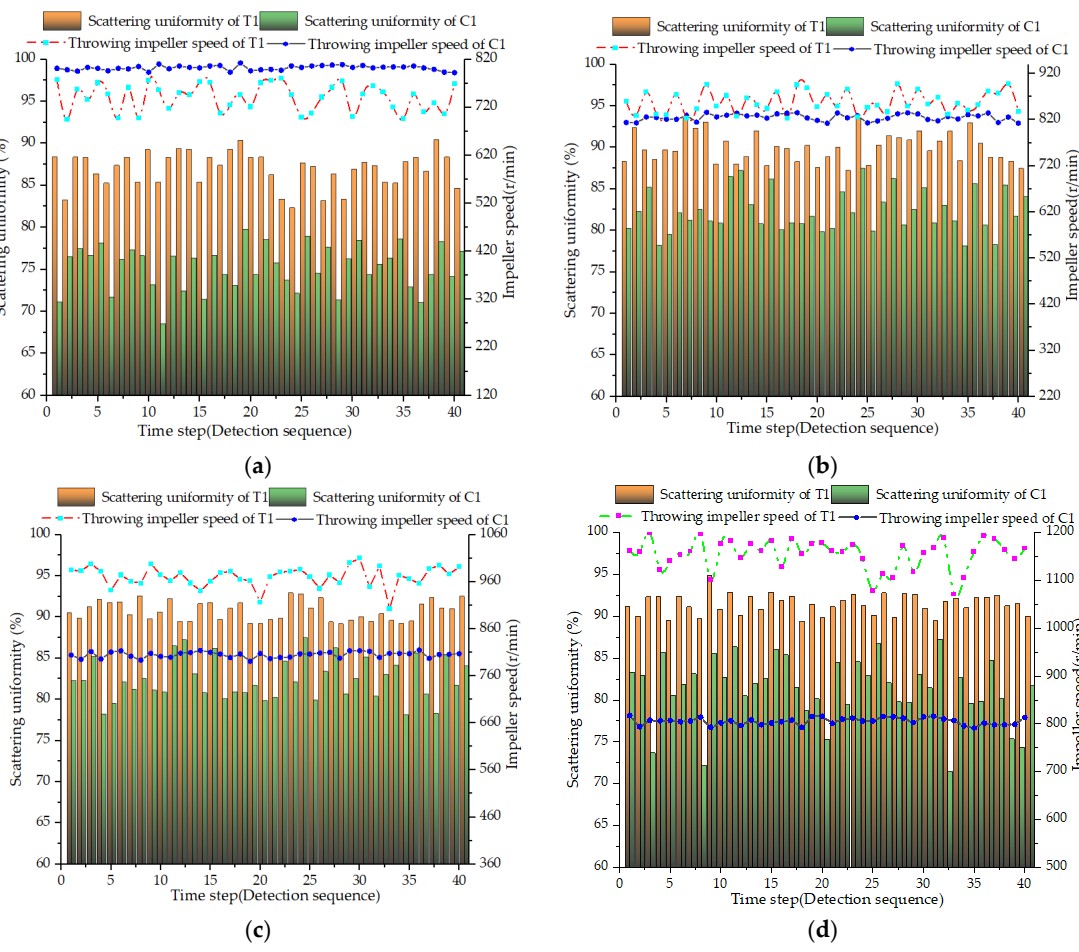

**Figure 10.** *Cont.*

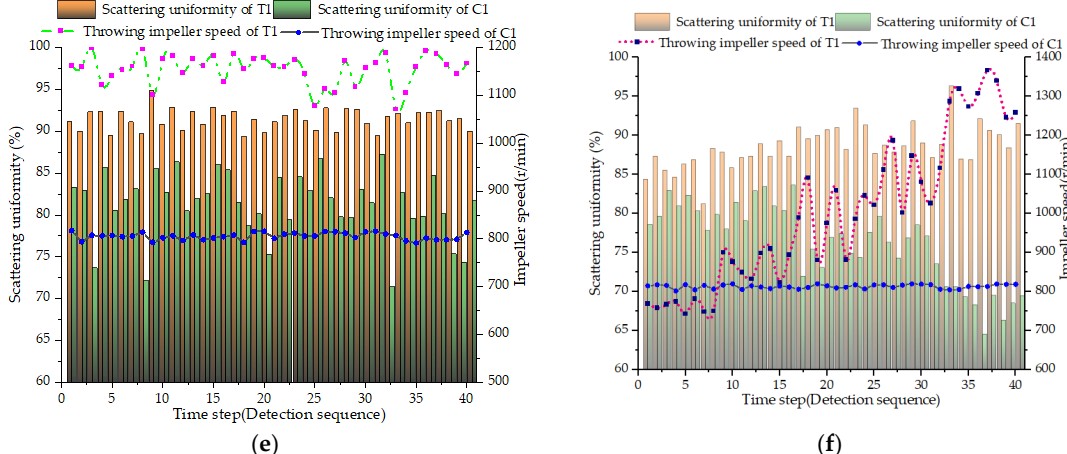

**Figure 10.** Changes in straw-scattering uniformity under T and C. (**a**) Changes in straw scattering uniformity in the test period at T1 and C1; (**b**) the changes in straw-scattering uniformity in the test period at T2 and C2; (**c**) the changes in straw-scattering uniformity in the test period at T34 and C3; (**d**) the changes in straw-scattering uniformity in the test period at T4 and C4; (**e**) the changes in straw-scattering uniformity in the test period at T5 and C5; (**f**) the changes in straw-scattering uniformity in the test period at T6 and C6.

Figure 10f shows the variation in straw-scattering uniformity at T6 and C6. It shows that the uniformity of straw scattering at the T6 level as 89.25%, and the average uniformity of straw scattering in each speed-keeping time period was 85.62%, 87.67%, 90.81%, 88.76% and 89.78%, respectively. The differences in average straw-scattering uniformity corresponding to T1, T2, T3, T4 and T5 were 1.30%, 2.31%, 0.06%, 2.73% and 0.72%, respectively, all of which were less than 3%, which indicated that at the T6 level, the automatic control system can sensitively adjust the rotating speed of the scattering impeller in real time, with the change in the forward speed (the amount of feed). At the C6 level, the test time was divided into five parts on average, and the corresponding average uniformity of the straw throwing was 80.35%, 81.25%, 75.24%, 76.13% and 68.37%, respectively; the difference between the different speed-keeping periods of T6 was 5.27%, 6.42%, 5.57%, 12.63% and 21.37%, respectively.

### 3.1.2. Influence of Different Levels on the Qualified Rate of Coverage

Figure 11a–e shows the changes in the qualified rate of straw coverage at 6 levels in the T and C treatments. The qualified rate of coverage corresponding to T1, T2, T3, T4 and T5 was 87.39%, 91.52%, 93.17%, 89.41% and 85.28%, respectively, with an average value of 85.28%. Under the five treatments of C, the corresponding average-coverage qualified rates were 74.23%, 75.51%, 82.80%, 78.26% and 75.69%, respectively, with an average of 77.30%. Under the two treatments of T and C, the average-coverage qualified rates and their fluctuations are random. The difference is that the qualified rate of the corresponding average coverage under the five T treatments was randomly in a high range, with a speed of 85.28~93.17%, and the variation range was approximately 8%. This variation range was large, which may be due to the different working conditions under different test treatments. However, the modified data showed that the automatic control system monitored the cutter-roller-shaft torque in real time. Under the condition of increasing feed rate, the impeller speed can be adjusted in the best turntable in real time, according to the torque change of the shaft, thus keeping the qualified rate of coverage stable, and in a high range. The average qualified rate of straw coverage under the five C treatments randomly increased at a low range of 74.23~82.80%, and the average value of the T treatment was 12.05% higher than that of the C treatment, which indicated that the impeller speed has different working conditions under different advancing speeds.

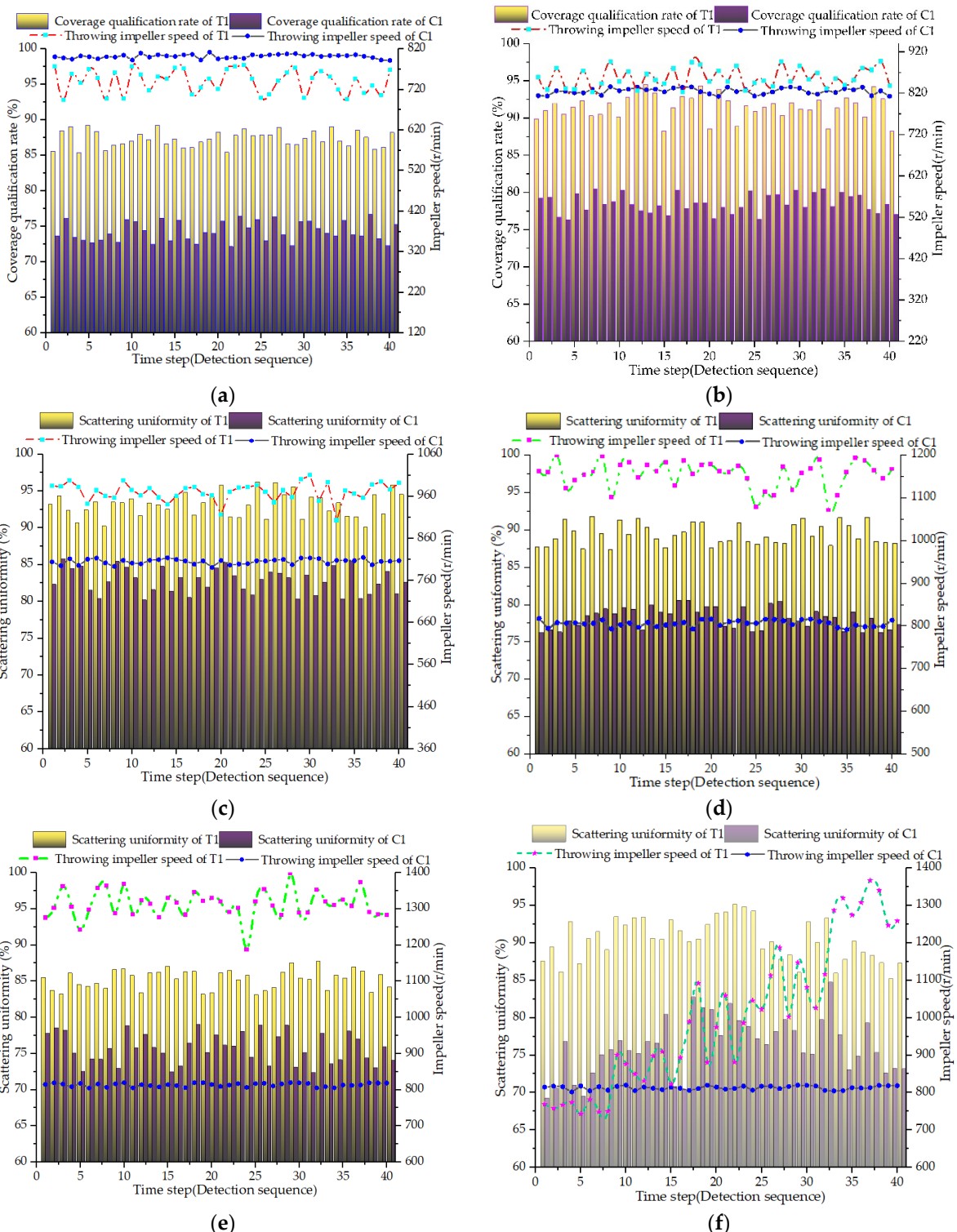

**Figure 11.** The changes in qualified rate of coverage in the test period at six levels under T and C. (**a**) The changes in qualified rate of coverage in the test period at T1 and C1; (**b**) the changes in qualified rate of coverage in the test period at T2 and C2; (**c**) the changes in qualified rate of coverage in the test period at T34 and C3; (**d**) the changes in qualified rate of coverage in the test period at T4 and C4; (**e**) the changes in qualified rate of coverage in the test period at T5 and C5; (**f**) the changes in qualified rate of coverage in the test period at T6 and C6.

Figure 11f shows the change in coverage qualification rate with the time step at T6 and C6 levels. The figure shows that the coverage qualification rate at the T6 level was 92.45%, and the average coverage qualification rate corresponding to each speed-keeping time period was 89.29%, 92.30%, 93.19%, 89.73% and 87.62% respectively, with an average of 90.43. The difference of the average qualified rate of coverage corresponding to T1, T2, T3, T4 and T5 was 1.90%, 0.78%, 0.02%, 0.32% and 2.34% respectively, all of which were less than 3%. It shows that at T6 level, the automatic control system can sensitively adjust impeller speed in real time with the change of the forward speed (the amount of feed), and make it rotate. Under the C6 level, the test time was divided into five parts on average, and the corresponding average qualified rates of coverage were 72.58%, 75.36%, 80.91%, 77.51% and 74.92%, respectively; and the difference between these and the T treatment was 16.71%, 16.94%, 12.28%, 12.22% and 12.72%, respectively.

## 4. Discussion

At present, no-till sowing machines are developing rapidly in China, and throwing uniformity of crushed straw is becoming a hot spot. In recent years, focusing on these two scenarios, Zhang et al. [25] adjusted the movement direction of straw by changing the spacing and angle of the guide blades at the tail of the test bed, to improve the uniformity of straw covering. The experimental result showed that the uniformity of straw scattering was 77.05%; Qin et al. [10] carried out numerical analysis and an experiment on straw crushing and the throwing impeller of straw crushing and fertilizing seeder. Results showed the uniformity of straw throwing was 88.1%, and the qualified rate was 90.2%. Gu et al. [3], focusing on the problem of conventional sowing machines being used in the main peanut-producing areas for many times, with a high production cost, carried out a large number of experiments on peanut no-till seeders working on full straw coverage, and important results were achieved. The uniformity of straw scattering was 83%. The similarities between this experiment and these studies are as follows: the crushing and conveying mechanism of the straw in the "initial stage" and "intermediate stage" of the test bed and the movement track of straw are similar to those of these two studies, but the differences are as follows: Zhang et al. [25] added guide vanes to the tail of the machine to adjust the movement direction of the crushed straw, in order to increase the coverage uniformity, but the guide vanes still need to be manually adjusted. There is no automatic adjustment of the impeller speed according to the amount of straw; Qin [10] and others have determined the key parameters of the straw-crushing-and-throwing impeller by means of theoretical analysis, a simulation optimization test, and structural optimization, etc., but still cannot adjust the throwing effect in real time, according to the amount of straw and the movement speed of machines and tools, and still cannot adapt to the feeding amount of the straw with a large fluctuation. However, Gu [3] used the method of changing the fan-speed parameters to influence the rotation speed of the straw-crushing-and-throwing impeller, and could not adjust the throwing effect in real time, according to the amount of straw and the moving speed of the machines. From the results of straw-throwing uniformity, the throwing uniformity of the straw was 12.6% higher than that of Zhang [25], and 1.55% higher than that of Qin [10] and Gu [10]. From the results of coverage uniformity, the difference between the results of this experiment and that of Qin et al. [10] is 0.85%, which is basically the same.

Scattered crushed-straw groups have disordered and chaotic. Therefore, although the feasibility and accuracy of trying to adopt the above methods have been verified in field experiments, but there is still a certain gap with the ideal goal. This study added an automatic speed control system of the throwing impeller, which effectively solved the problems of low-straw-throwing uniformity and the qualified rate of straw coverage caused by the real-time change in the feed amount of crushed straw caused by the unstable forward speed of the machines (or test beds) or the large difference in straw quality per unit area in the test field. Based on the important practical demands of this problem, this experiment innovatively realized the automatic control of the impeller speed at different advancing

speeds. Therefore, it is rare around the world to effectively solve the "last key link" of no-till sowing. Therefore, this research has important innovative value.

Although the research has innovations, there are some limitations:

(1) Although the impeller speed is changed according to the real-time change of the torque of the cutter roller shaft, smashing knives (groups) of the cutter roller shaft will hit the ground during the rotation, which brings uncertainty and certain test errors to the real-time monitoring of the torque signal. In the next step, more sensors and intelligent algorithms will be added to the system to reduce the torque of the knives entering the soil.

(2) The system mainly converts the torque signal into the signal of the straw amount in real time, but the torque signals of different crops or straws of the same crop are different under different moisture-content conditions. The next step will be to expand the test scale, obtain a large number of field test data, explore the law and mathematical model of moisture content on the torque signal, and introduce this into the automatic control system to increase the adaptability to moisture content.

### 5. Conclusions

(1) The average uniformity of straw scattering corresponding to T1, T2, T3, T4 and T5 was 86.97%, 89.98%, 90.75%, 91.49% and 89.06%, respectively, and the average value was 89.65%. The average evenness of straw scattering under the five treatments of C was 75.23%, 82.12%, 82.42%, 76.27% and 71.41%, respectively. The average evenness of straw scattering and its fluctuation under the five treatments of T and C showed the characteristics of increasing the advancing speed at first and decreasing at random.

The average uniformity of straw scattering in the T6 speed-keeping time period was 85.62%, 87.67%, 90.81%, 88.76% and 89.78%, respectively, and the difference between the average uniformity of the straw scattering in T1, T2, T3, T4 and T5 was 1.30%, 2.31%, 0.06% and 2.73%, respectively. The automatic control system can sensitively adjust the impeller speed in real time with the change in forward speed (the amount of feed), and make the impeller speed function within a reasonable range, so that the uniformity of the straw throwing is always higher than 86.12%. At C6 level, the test time was divided into five parts on average, and the corresponding average uniformity of the straw throwing was 80.35%, 81.25%, 75.24%, 76.13% and 68.37%, respectively; the difference between the different speed-keeping periods of T6 was 5.27%, 6.42%, 5.57%, 12.63% and 21.37%, respectively.

(2) The average qualified rate of coverage corresponding to T1, T2, T3, T4 and T5 was 87.39%, 91.52%, 93.17%, 89.41% and 85.28%, respectively, with an average of 89.35%. The average qualified rate of coverage corresponding to the five treatments of C was 74.23%, 75.51% and 85.28%, respectively. The average qualified rate of coverage corresponding to each time period of T6 speed maintenance was 89.29%, 92.30%, 93.19%, 89.73% and 87.62%, with an average of 90.43%. The difference in the qualified rate of average coverage corresponding to T1, T2, T3, T4 and T5 was 1.90%, 0.78% and 0.02, respectively. In the experiment, the average qualified rate of coverage in C6 was 72.58%, 75.36%, 80.91%, 77.51% and 74.92%, and the difference between these and the T treatment was 16.71%, 16.94%, 12.28%, 12.22% and 12.70%, respectively, which indicates that the automatic control system could improve the qualified rate of coverage.

**Author Contributions:** Conceptualization, F.G.; methodology, H.Y.; software, H.Y.; validation, B.W.; formal analysis, H.W.; investigation, F.W.; resources, H.W.; data curation, H.W.; writing—original draft preparation, F.W.; writing—review and editing, B.W.; visualization, F.W.; supervision, Z.H.; project administration, H.W.; funding acquisition, B.W. All authors have read and agreed to the published version of the manuscript.

**Funding:** This research was funded by the following project funds: The Natural Science Foundation of Jiangsu Province (grant number BK20221187); National Peanut Industry Technology System, (grant number CARS-13).

**Institutional Review Board Statement:** Not applicable.

**Informed Consent Statement:** Not applicable.

**Data Availability Statement:** The datasets used and/or analyzed during the current study are available from the corresponding author upon reasonable request.

**Acknowledgments:** The authors would like to thank the teacher and supervisor for their advice and help during the experiments. We also appreciate the editor and anonymous reviewers for their valuable suggestions for improving this paper.

**Conflicts of Interest:** The authors declare no conflict of interest.

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
