# Peer review of "Experimental Analysis and Evaluation of Automatic Control System for Evenly Scattering Crushed Straw"

_agriculture, doi:10.3390/agriculture13030679_

Round 1

Reviewer 1 Report

The article is novel  and the data shown in the graphs agree with the data mentioned to have been obtained in field tests.

• The mathematical models are consistent with what is presented.

• With respect to the structure of the article, everything follows the regulations, but there is a section on "patents" where there is no information related to the patent, so it would be prudent to incorporate the information or remove that section.

• In the conclusions section, they are presented in a list format. It would be more appropriate to write them in paragraphs without numbering them. In one of the conclusions, it is cited "Therefore, this test was innovative and scientific." Although it is true that this article is innovative, it is not necessary to include those comments in the conclusions.

• It would be prudent to polish the article's wording a bit more since it contains some difficult-to-understand words and some typos such as the lack of spaces in some sentences.

• There are few references, and it is suggested to increase their number.

• It is necessary to expand the context of the proposal for the automatic control system in the introduction.

Reviewer 2 Report

lines 156-157 please give reference to equation number 

lines  289-290 please give reference to equation number 

chapter 3 - please explain more clear -  the scientific contributions and the  results  - compared with the literature, on:

- the modelling

- the simulation

- the experimental validation of the model 

paragraph 3.1.1 - please give reference and explanations to fig 10

paragraph 3.1.2 - please give reference and explanations to figures -  lines 378-385 

Reviewer 3 Report

Experimental analysis and evaluation of automatic control system for evenly scattering crushed straw

·         The paper is generally prepared in a standard format with sound literary and technical presentation. It is well written in clear and concise manner. The figures and schemes are properly presented.

·         The paper is a good report on a study to reduce the problems of poor uniformity of straw spreading and low rate of coverage in China.

However, the authors should make the following corrections to make the paper acceptable for further consideration and subsequent final acceptance:

Title: The title can be simplified as: Evaluation of automatic control system for evenly scattering of crushed straw

Abstract: The abstract is very clumsy. It does not adhere to the universally acceptable format of abstract writing - Introduction with problem statement, objectives, methodology, results & discussion and conclusion. Please rearrange your abstract in this format.

·         What do you mean by ‘qualified rate’?

·         Line 13: ‘Aiming at the problems’ should be changed to ‘Aiming at solving the problems’

·         Line 26: ‘Taking’ should be changed to ‘taking’

Materials and Methods:

Lines 127-131: This section should be recast to give clearly itemized information as intended

Line 137: change ‘output’ to ‘outputs’

Discussion

Given the elaborate materials and methods dedicated to this research, the discussion is comparatively shallow.

Line 390: ‘aiming at these two problems’ – No problems have been identified here. Or rather, change ‘problems’ to ‘scenarios’

Lines438-462: Instead of having a separate section for limitations, each limitation should be embedded in the body of discussion.

 Conclusion

·                    The conclusion section of your article is not concise. It should end with the summary of your thoughts and convey the larger implications of your study. It shouldn’t be used for another round of results and discussion.

Round 2

Reviewer 2 Report

the paper can be published